# Stimulative Effects of *Lupinus* sp. and *Melilotus albus* Underseed on the Photosynthetic Performance of Maize (*Zea mays*) in Two Intercropping Systems

Jaroslav Lang [1], Peter Váczi [2], Miloš Barták [2], Josef Hájek [2,*], Antonín Kintl [1], Barbora Zikmundová [1] and Jakub Elbl [3]

1 Department of Agrotechnics, Research Institute for Fodder Crops, Zahradní 1, 664 41 Troubsko, Czech Republic
2 Laboratory of Photosynthetic Processes, Department of Experimental Biology, Faculty of Science, Masaryk University, Kamenice 5, 625 00 Brno, Czech Republic
3 Department of Agrosystems and Bioclimatology, Faculty of Agrisciences, Mendel University in Brno, Zemědělská 1, 613 00 Brno, Czech Republic
* Correspondence: jhajek@sci.muni.cz

**Abstract:** In order to evaluate influential mechanisms for photosynthetic processes on the yields of an intercropping system composed of maize (*Zea mays*), *Lupinus* sp. and *Melilotus albus*, three treatments were designed and conducted in southern Moravia (Czech Republic) in the form of agronomy trials. The treatments included sole maize (SM), maize with *Lupinus* sp. (ML) and maize with field melilot (MM). The photosynthetic processes of *Zea mays* were monitored using several chlorophyll fluorescence techniques on the three treatments for 20 days in the late summer season. An analysis of fast chlorophyll fluorescence transients (OJIP) showed that the capacity of photochemical photosynthetic reactions in photosystem II ($F_V/F_M$), as well as the photosynthetic electron transport rate ($ET_0/RC$), declined in response to a four-day episode of extremely warm days with full sunshine. Similarly, the performance index (PI), an indicator of general plant vitality, declined. The episode activated protective mechanisms in photosystem II (PSII), which resulted in an increase of thermal dissipation. For the majority of *Z. mays* photosynthetic parameters, their values decreased for particular treatments in the following order: MM, ML, SM. The MM and ML intercropping systems had a positive effect on the primary photosynthetic parameters in *Z. mays*.

**Keywords:** fast chlorophyll fluorescence transients; light response curves; primary photosynthetic processes; white sweetclover

## 1. Introduction

Maize (*Zea mays*) is among the most cultivated crops in central Europe. Recently, the most popular agricultural practice has been to grow maize in monocultures in large areas of agricultural land. That is why the cultivation of maize in Europe faces some challenges regarding the low biodiversity of agricultural ecosystems. Recently, the approach of intercropping maize with other crops has become an important issue, since it increases the biodiversity of the fields with implications for crop production, as shown by [1]. The system of intercropping of maize with white sweetclover and white lupin has been studied recently by [2,3]. According to Turkington et al. (1978) [4], white sweetclover is one of the alternative legumes for biogas production. The species is mainly used as a fodder crop, a soil-improving crop on less fertile soils and a green manure crop. Kadaňková et al. (2019) [5] and Kintl et al. (2022) [6] evaluated the potential of white sweetclover for intercropping systems.

Simultaneous intercropping has recently become a common agronomy practice to promote the production of a target crop species. The two approaches affect the agronomic performance of the target crop, dry matter production, partitioning and grain yield [7]. Among the combinations most commonly used in intercropping are cereals and legumes,

with the association of maize [8,9]. The potential use of the *Fabaceae* family for agricultural soil sustainability has been reviewed in several studies [10–12] that focused the effects of stand density, and the intercropping system designs on crop production.

Chlorophyll fluorescence parameters are used routinely to indicate limitations in the photosynthetic apparatus, photosystem II (PSII) in particular, in response to a variety of biotic and abiotic stressors. In crops, special attention is paid to drought stress [13,14], its effect on effective quantum yield of PSII and photosynthetic parameters.

Among the recently-used chlorophyll fluorescence techniques used in crop research, fast chlorophyll fluorescence transients (OJIP) have been increasingly applied since they are non-invasive, fast and allow us to collect and analyze large datasets in a reasonable timeframe. The OJIPs are measured in crops in order to evaluate OJIP characteristics [15] and relate them to the particular driving factors affecting the primary photosynthetic processes in the leaves. In this concept, increased evidence for the negative effects of drought stress [16,17], salinity stress [18,19], high-temperature stress [20] and heavy metal stress [21] have accumulated in the last few decades. In this paper, we tested the hypothesis that the two intercropping species used in our study (legumes: *Lupinus* sp., *Melilotus albus*) would affect the microclimate of the stand and improve soil quality—nitrogen availability in particular. In our concept, these changes would lead to an increase in the photosynthetic performance of the target species (*Zea mays*) and, consequently, increased biomass production.

## 2. Materials and Methods

### 2.1. Field Experiment Design

The experimental plot covered a total area of 100 m$^2$. The experiment was based on the randomization method in three replications (on individual 6 × 20 m plots). The selected crops were maize (*Zea mays* L., FAO 230–240), lupin (*Lupinus* sp.) and white sweetclover (*Melilotus albus* Medik., Meba). For details and the experimental design, see [3]. Before sowing, mixed NPK (10:26:26) fertilizer was applied (HOKR, spol. s r.o., Pardubice, Czech Republic) at 300 kg/ha.

(1).　Monoculture A: maize (*Zea mays*), the treatment abbreviated SM

Seventy-five thousand units/ha of selected maize seed were sown in 0.375 m rows using a Kinze 3500 precision vacuum seeder "interplant system" (Kinze Manufacturing, Williamsburg, IA, USA).

(2).　Mixed maize and white sweetclover (*Melilotus albus*), the treatment abbreviated MM

The combination of maize and white sweetclover was sown on the same date as the maize monoculture, again using a Kinze 3500 precision vacuum seeder (Kinze Manufacturing, Williamsburg, IA, USA). The maize was sown in a row at a distance of 0.375 m and the white sweetclover was sown in a strip 0.375 m wide on each side, 0.375 m from the nearest maize row. Both crops were sown at the same time. The sowing rate of each crop was 75 thousand grains/ha, and the total number of individuals (maize + white sweetclover) was thus 150 thousand grains/ha.

(3).　Mixed maize and white lupin (*Lupinus* sp.), the treatment abbreviated ML

The combination of maize and lupin was sown on the same date using a Kinze 3500 precision vacuum seeder (Kinze Manufacturing, Williamsburg, IA, USA). Individual grains of maize and lupin were sown simultaneously in rows at a distance of 0.375 m. The sowing rate of each crop was 75 thousand grains/ha, bringing the total number of individuals (maize + lupin) to 150 thousand grains/ha.

### 2.2. Soil Characteristics

The experimental plot (Figure 1) is located close to Agricultural Research Ltd. in Troubsko (49.1709° N, 16.4916° E, Figure 2). In 2018, winter wheat was grown as a preceding crop on the plot. The trial was established in 2019. According to agroecological classification, the plot is located in a region typical for the cultivation of root crops (e.g., sugar beets) in a

mildly warm and mildly dry climatic zone at an altitude of 290 m above sea level, with a mean annual temperature of 8.95 °C and a long-term total annual precipitation of 525.6 mm (the values correspond to the climatic norm of 1981–2010). The geological bedrock is composed of loess and loess loam of the Bohemian Massif, and the soil type is Haplic Luvisol. Basic information about the characteristics of the arable land on the experimental plot can be found in Table 1. The meteorological and climatological parameters are shown in Figure 3.

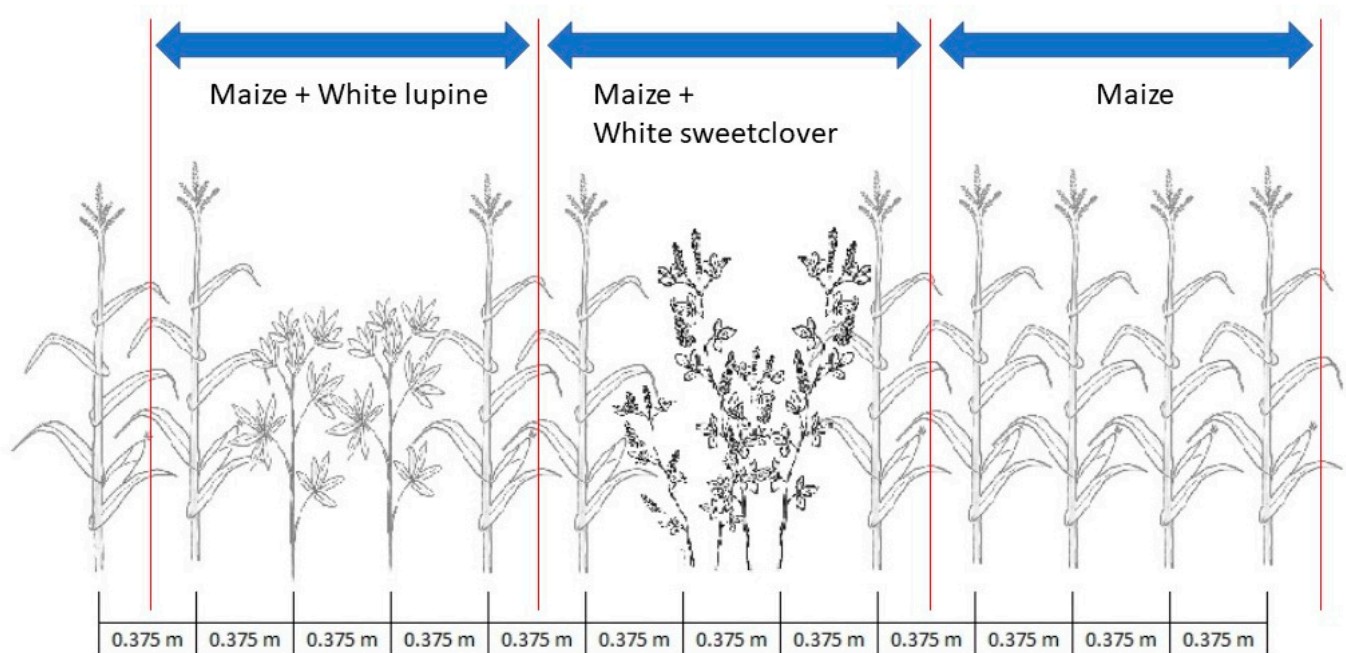

**Figure 1.** Scheme of intercropping system used in the study. The treatments are abbreviated: ML (maize and *Lupinus* sp.), MM (maize and *Melilotus albus*), and SM (maize).

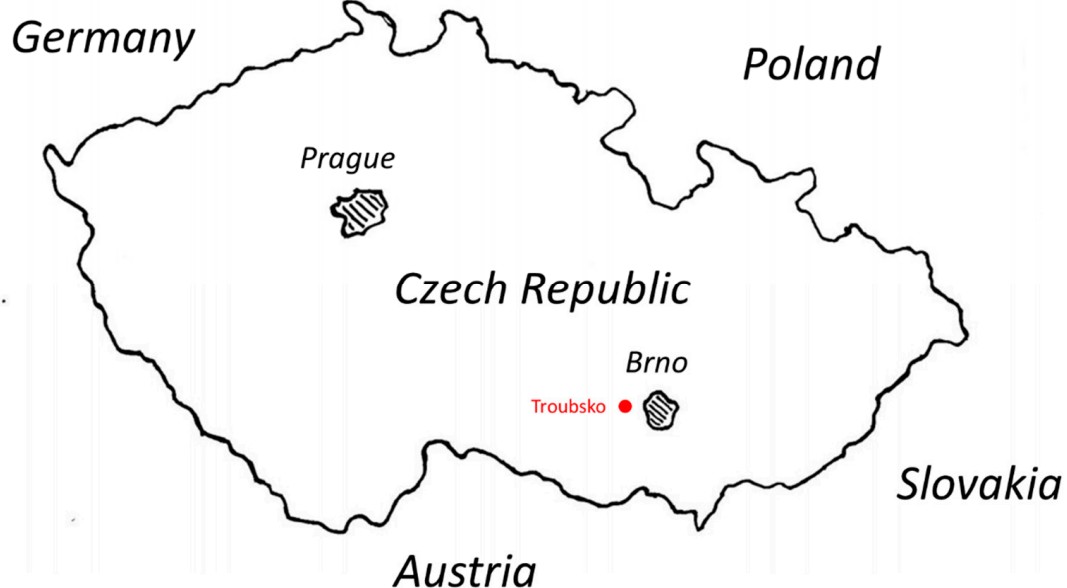

**Figure 2.** Localization of the Troubsko experimental field station in CZE (Czech Republic).

**Table 1.** Characteristics of the arable soil from the experimental plot and the average contents of plant available nutrients. *Note*: Mean of measured values (*n* = 3) is shown ± standard deviation (SD).

| Sample | Soil Reaction (pH) | Plant Available Nutrient Content (mg/kg) | | | | |
|---|---|---|---|---|---|---|
| | | $N_{an}$-100% | K | Mg | P | Ca |
| Arable Soil | 7.1 ± 0.2 | 23.9 ± 5.3 | 191 ± 34.2 | 153 ± 5.1 | 58.3 ± 5.1 | 5968 ± 556 |

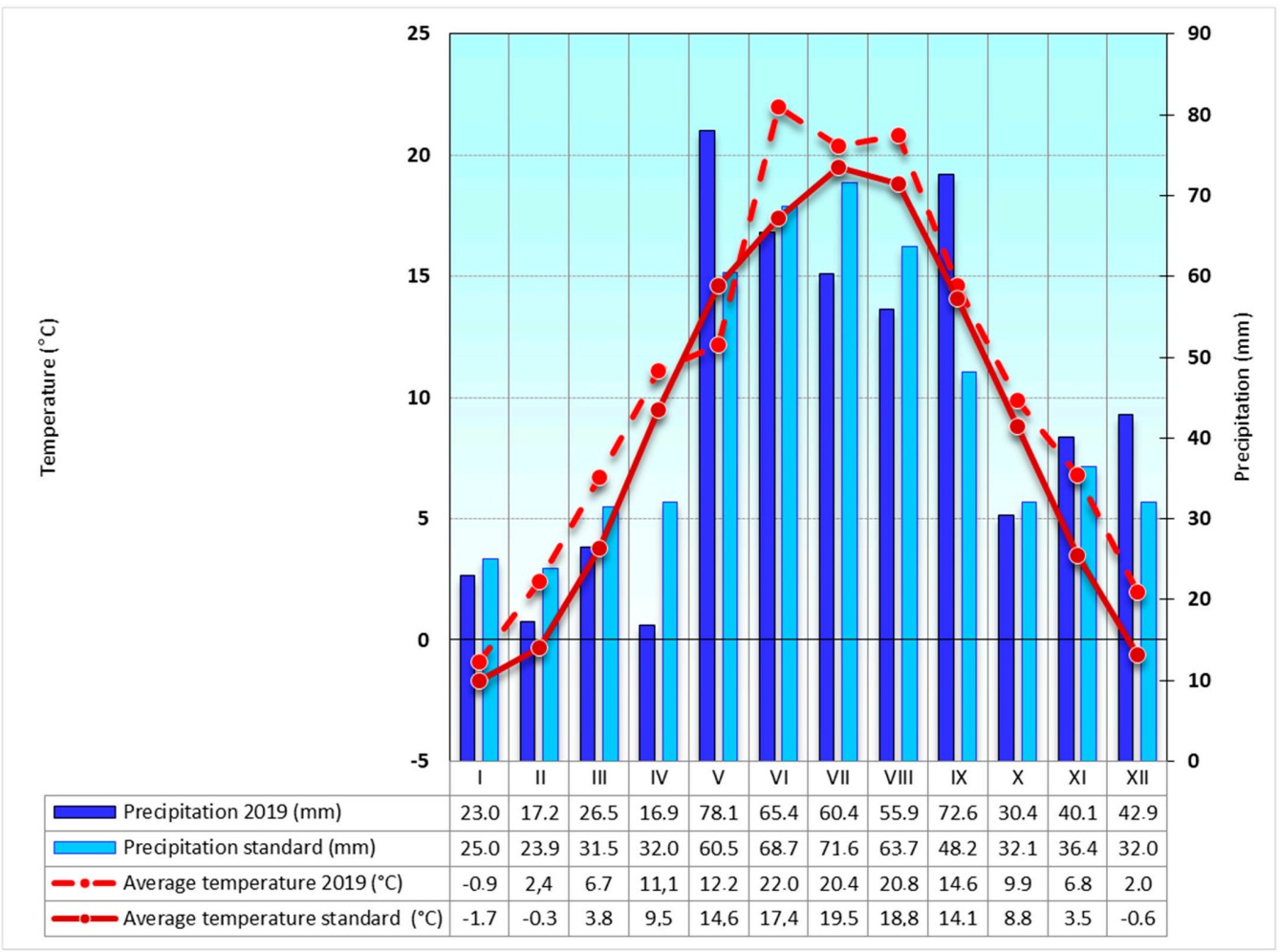

| | I | II | III | IV | V | VI | VII | VIII | IX | X | XI | XII |
|---|---|---|---|---|---|---|---|---|---|---|---|---|
| Precipitation 2019 (mm) | 23.0 | 17.2 | 26.5 | 16.9 | 78.1 | 65.4 | 60.4 | 55.9 | 72.6 | 30.4 | 40.1 | 42.9 |
| Precipitation standard (mm) | 25.0 | 23.9 | 31.5 | 32.0 | 60.5 | 68.7 | 71.6 | 63.7 | 48.2 | 32.1 | 36.4 | 32.0 |
| Average temperature 2019 (°C) | -0.9 | 2,4 | 6.7 | 11,1 | 12.2 | 22.0 | 20.4 | 20.8 | 14.6 | 9.9 | 6.8 | 2.0 |
| Average temperature standard (°C) | -1.7 | -0.3 | 3.8 | 9,5 | 14,6 | 17,4 | 19.5 | 18,8 | 14.1 | 8.8 | 3.5 | -0.6 |

**Figure 3.** Weather conditions at the Troubsko field experimental station, 2019—average monthly temperatures and mean annual precipitation amounts for the long term are standard for 1981–2010.

*2.3. Local Climate*

The Troubsko site is located southwest of Brno, in the northern part of the Pannonian thermophytic zone. It belongs to the beet production area, with an altitude of 270 m and an average annual temperature of 8.6 °C, 14.8 °C in vegetation (April–September). The temperature and precipitation data for the Troubsko experimental station were taken from 1961 to 1990 by a standard meteorological station run by the Czech Hydrometeorological Institute on the Agricultural Research Ltd. site in Troubsko, Czech Republic (49.164128° N, 16.511900° E).

The temperature and precipitation data show that 2019 was exceptionally warm, and precipitation reached a long-term average value (Figure 3). High air temperature was reached mainly in the first part of the year and during the growing season (April–September) as well. Despite the normal rainfall pattern, a lack of moisture was apparent in the fields because of increased transpiration and evapotranspiration due to the high air temperature (evaluated according to WHO methodology by [22].

### 2.4. Measurements of Microclimate

For microclimate measurements, a standard weather station (EMS, Brno, Czech Republic) monitoring air temperature and humidity was used, along with a photosynthetically active radiation (PAR) incident on the upper canopy (height of 2 m) in 10 min intervals. Based on the data, daily courses of air, T, RH and PAR were plotted and analysed.

### 2.5. Measurements of OJIPs and OJIP-Derived Parameters

To determine changes in photosynthesis at the level of photosystem 2, fast chlorophyll fluorescence kinetics (OJIP) were measured using a handy fluorometer FluorPen (FP-100, Photon Systems Instruments, Drásov, Czech Republic) in the morning (typically 8:30 to 11:00 CEST) at the beginning, and on the 2nd, 5th, 7th, 9th, 12th, 14th, 16th, 19th, 21st and 23rd days of the monitored period. The measurements were done on the third fully matured upper leaf (counted from the top), determined for all variants at the beginning, and remained the same throughout the measuring period of the experiment. For each plant species and the day of measurements, 5 replicates of OJIP curves were measured on different leaves. Measuring protocol started with a predarkening period of 15 min in the darkening clip, then OJIP kinetics were recorded (2 s, standard FluorPen OJIP protocol). The measured kinetics were analyzed, and OJIP-derived parameters were calculated using FluorPen software (v. 1.1, Photon Systems Instruments, Drásov, Czech Republic) according to equations published by [15]. The overview of the parameters used in the study are presented in Table 2.

**Table 2.** OJIP-derived parameters used to study changes in photosystem II photochemistry.

| Abbrev. | Formula/Equations | Explanation |
|---|---|---|
| $F_V/F_M$ | | Maximal quantum yield of PSII fluorescence |
| $V_J$ | $V_J = (F_J - F_0)/(F_M - F_0)$ | Relative variable fluorescence at the J-step |
| $PI_{ABS}$ | $PI_{ABS} = (RC/ABS) \cdot [\varphi_{P0}/(1 - \varphi_{P0})] \cdot [\psi_0/(1 - \psi_0)]$ | Performance index (potential) for energy conservation from exciton to the reduction of intersystem electron acceptors |
| $ET_0/RC$ | $ET_0/RC = M_0 \cdot (1/V_J) \cdot \psi_0$ | Electron transport flux (further than $Q_A$) per RC |
| $DI_0/RC$ | $DI_0/RC = (ABS/RC) - (TR_0/RC)$ | The flux of dissipated excitation energy at time 0 |
| QY_L1 | | Equivalent to effective quantum yield of PSII ($\Phi_{PSII}$) |

In order to determine the main driving factors limiting primary photosynthetic processes (OJIP-derived parameters), correlations between selected OJIP-derived parameters ($F_V/F_M$, $PI_{ABS}$, $ET_0/RC$, and $DI_0/RC$-dependent variables) and the following independent variables were evaluated by STATISTICA package: (1) mean air temperature of the day before chlorophyll fluorescence measurements, (2) mean PAR of the day before measurements, (3) mean RH of the day before measurement, (4) maximum air temperature of the day before measurements, (5) maximum RH of the day before measurements, (6) minimum air temperature of the day before measurements, (7) minimum RH of the day before measurements, (8) mean air temperature of the period of measurement (7:00 to 11:00 a.m. of particular day), (9) mean PAR of the period of measurement, (10) mean RH of the period of measurement, (11) maximum air temperature of the period of measurement, (12) maximum PAR of the period of measurement, (13) maximum RH of the period of measurement, (14) minimum air temperature of the period of measurement, (15) minimum PAR of the period of measurement and (16) minimum RH of the period of measurement.

### 2.6. Light Response Curve Analysis

To determine changes in the rate of photosynthesis to irradiance, the light response curves were measured. The method is based on the successive effective quantum yield measurement of the sample when exposed to a stepwise increase of light intensity. The measurements were taken using a predefined protocol by the FluorPen fluorometer. The

effective quantum yields were measured after each 60 s period of exposition by light intensities of 10, 20, 50, and 100 μmol $m^{-2}$ $s^{-1}$. The measurements were done each measuring day (2nd, 5th, 7th, 9th, 12th, 14th, 16th, 19th, 21st and 23rd) of the monitored period. For each treatment, i.e., for (1) maize (SM), (2) maize and white sweetclover (MM) and (3) maize and white lupin (ML), the best and worst curves were selected (considering the highest and lowest $ETR_{max}$ values reached s) and the alpha parameter, i.e., maximum slope of the ETR to PAR relationship found at low light intensities, was also evaluated.

### 2.7. Biomass Determination

Above-ground biomass (dry mass) was harvested at the end of the experiment (mid-August) and estimated for *Z. mays* grown in the three systems: SM, MM, ML. Additionally, above-ground biomass of *Melilotus albus* in the MM intercropping system and *Lupinus* sp. grown in the ML intercropping system were evaluated.

### 2.8. Statistical Analysis

To calculate the statistical significance of the differences found in chlorophyll fluorescence parameters as dependent on the day of measurement and the intercropping system, one-way ANOVA was used (STATISTICA vs. 14, StatSoft-TIBCO Software Inc., Prague, Czech Republic).

## 3. Results

### 3.1. Local Microclimate

Air temperature varied in the experimental period according to the prevailing weather. In the period from 19 July to 27 July, warm days showed daily maxima ranges between 31.2 and 37.3 °C. The daily minima varied within the range of 12.5 to 14.1 °C. Partly to fully sunny days were typical of this particular period, especially in the subperiod of 23 July to 26 July. Overcast weather recorded on 27 July led to a decrease in air temperature, showing a maximum of 22.1 °C. Then, in the period of 28 July to 2 August, intermediate weather was typical in the form of partly sunny days with slightly decreasing daily maxima for PAR and air temperature. At the end of this period, another overcast day appeared on 3 August, with daily minimum/maximum values of 13.2/20.0 °C. The third period (4–9 July) was marked by gradual increases in air temperature, with daily maxima above 25 °C and minima ranging between 10.5 and 15.1 °C. Generally, the whole experimental period saw warm weather with low values of air relative humidity (RH minima below 23% in hot sunny days, data not shown).

### 3.2. OJIP-Derived Parameters of Photosynthesis

The values of the maximal quantum yield of PSII fluorescence ($F_V/F_M$, Figure 4) varied from 0.786, i.e., the minimum recorded on the 9th day for the SM system (*Zea mays* alone), to a maximum of 0.838 recorded in the 2nd day for the MM intercropping system (*Zea mays* with *Melilotus albus*). All monitored treatments followed a similar trend with a decrease from the beginning (day 0) to 9th day, and a subsequent gradual rise until the end of the monitoring period. Throughout the monitoring period, $F_V/F_M$ recorded for the ML treatment (*Zea mays* with *Lupinus* sp.) showed somewhat higher values than the other two.

Double-normalized fluorescence at J-step ($V_J$, Figure 4) showed variation between 0.445, a minimum that was found on the 23rd day of the MM intercropping system, to a maximum of 0.613 that was recorded on the 9th day for *Zea mays* alone (SM). The $V_J$ time course showed an initial plateau until the 5th day, with a subsequent rise which followed until the 9th day, and then a decrease until the 16th day. More or less constant $V_J$ was found at the end of the monitored period. The three treatments, i.e., the SM, MM and ML systems, followed the same time courses.

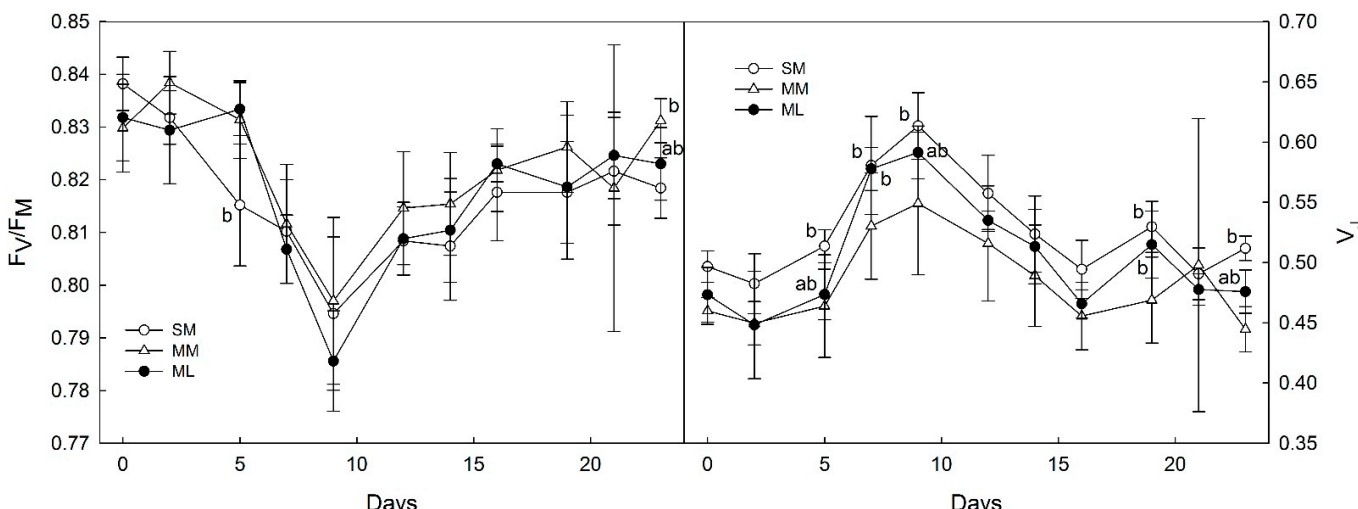

**Figure 4.** Time courses of $F_V/F_M$ (**left**) and $V_J$ (**right**). *Note*: *Zea mays* (SM), *Zea mays* + *Lupinus* sp. (ML), *Zea mays* + *Melilotus albus* (MM). Data points represent means calculated from five replicates $\pm$ standard deviations. Statistically significant differences are indicated by the letters (a, b). Most common (a) is not shown due to too numerous appearances. Only (ab) and (b) are shown.

The performance index ($PI_{ABS}$, Figure 5) showed similar time courses as $F_V/F_M$ within the monitored period. An initial plateau was typical of the time course, followed by a decrease found between the 5th and 9th day and a subsequent increase until the 16th day. The maximum of 2.876 was recorded on the 2nd day for the MM treatment, and a minimum of 0.861 showed for *Zea mays* alone on the 5th day. The photosynthetic electron transport flux per RC ($ET_0/RC$, Figure 5) varied from a minimum of 0.894, recorded on the 9th day for *Zea mays* alone, to a maximum of 1.069, found on the 2nd day for the ML intercropping system. The ET0/RC time course exhibited a slight initial rise until the 5th day and was then followed by a decrease until the 7th day. Another rise in $ET_0/RC$ was found until the 12th day, then $ET_0/RC$ decreased until the 16th day, followed by a final slight rise.

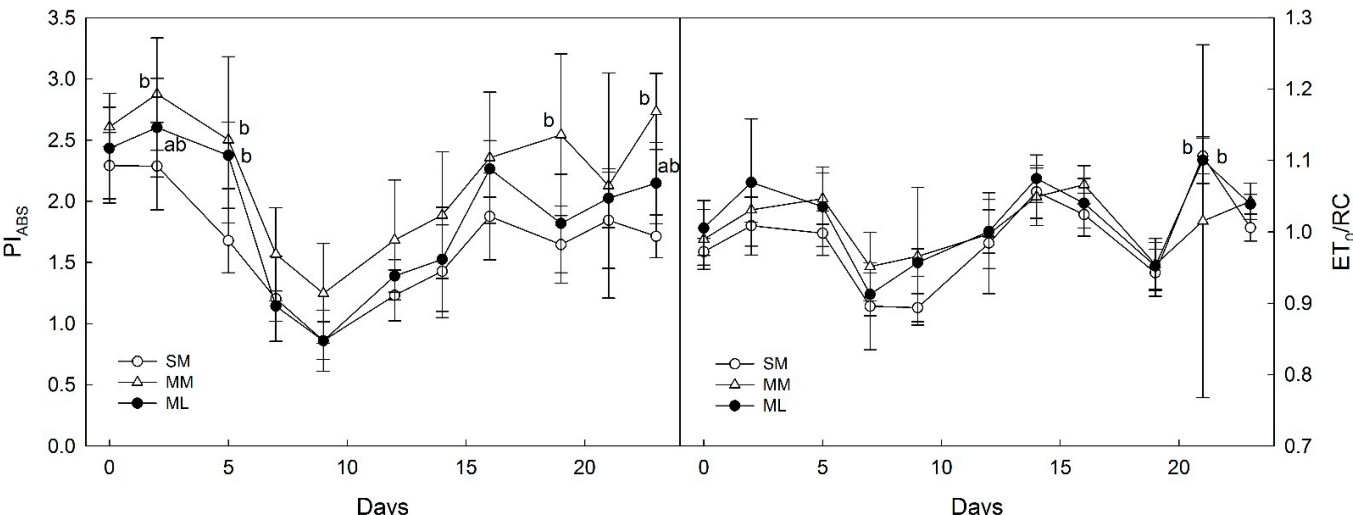

**Figure 5.** Time courses of $PI_{ABS}$ (**left**) and $ET_0/RC$ (**right**). *Note*: *Zea mays* (SM), *Zea mays* + *Lupinus* sp. (ML), *Zea mays* + *Melilotus albus* (MM). Data points represent means calculated from five replicates $\pm$ standard deviations. Statistically significant differences are indicated by the letters (a, b). Most common (a) is not shown due to too numerous appearances. Only (ab) and (b) are shown.

The flux of dissipated excitation energy per RC ($DI_0/RC$, Figure 6) showed in the initial plateau with the overall minimum found on the 2nd day for the MM intercropping system, followed by a rise to the maximum of 0.642 for the ML system (0.602 for SM and

0.549 for MM) on the 9th day. Then, the values gradually decreased until the end of the monitored period, reaching a minimum close to the initial values. The effective quantum yield ($\Phi_{PSII}$, OY_L1, Figure 6) varied between 0.570 (the minimum for the SM treatment on the 9th day) and 0.727 (the maximum for the MM on the 2nd day). For the three treatments, $\Phi_{PSII}$ showed a decrease from the beginning to the 9th day, followed by an increase to initial values found on the 14th (SM, ML) to the 16th day (MM).

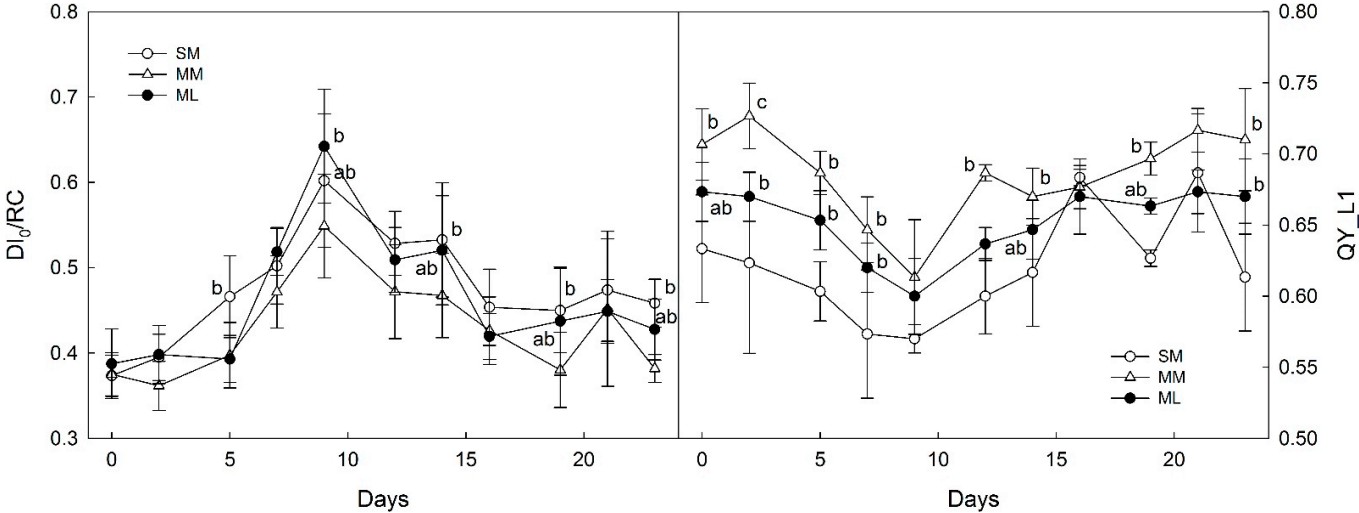

**Figure 6.** Time courses of $DI_0/RC$ (**left**) and $\Phi_{PSII}$ (**right**). *Note*: *Zea mays* (SM), *Zea mays + Lupinus* sp. (ML), *Zea mays + Melilotus albus* (MM). Data points represent means calculated from five replicates ± standard deviations. Statistically significant differences are indicated by the letters (a, b). Most common (a) is not shown due to too numerous appearances. Only (ab) and (b, c) are shown. Parameter QY_L1 refers to effective quantum yield of photosystem II.

Air temperature and PAR had the most significant effects on the OJIP-derived chlorophyll fluorescence parameters related to primary photochemistry ($F_V/F_M$, $PI_{ABS}$, $ET_0/RC$, and $DI_0/RC$) as seen from Table 3. Typically, temperature and PAR on the day preceding the day of measurement had a stronger effect than those evaluated for the period of the measurement. Relative air humidity had lower effects, if any, on the above-mentioned chlorophyll fluorescence parameters.

ETR curves recorded on the days with different microclimatological characteristics showed a contrasting course and shape. The best ETR curves (those having the largest value of $ETR_{max}$) were recorded on 29 or 31 July, i.e., day 12 and 14. The microclimate of the days was typical of $PAR_{max}$ ranging 900–1000 µmol m$^{-2}$ s$^{-1}$ (see Figure 7) and daily mean temperature of 25 °C. The worst ETR curves were recorded on days 19 to 23 which were typical by comparable $PAR_{max}$ but lower daily means of temperature. The latter ones were typical of a reduction of $ETR_{max}$ values found at 100 µmol m$^{-2}$ s$^{-1}$. For the first category of days, the best ETR curve showing the highest ETR values was found and presented in Figure 8. Photosynthetic performance based on $ETR_{max}$ values was found to be best in the intercropping system of maize with *Melilotus albus*, followed by the maize with *Lupinus* sp. and maize alone. For the second category of days, the difference between $ETR_{max}$ found for particular intercropping systems was smaller, however, the values found for maize were lower than for the other two intercropping systems.

**Table 3.** Correlation between climatic parameters and OJIP-derived chlorophyll fluorescence parameters related to PSII primary photosynthetic processes. Red numbers (r values) indicate statistically significant difference. Climatic parameters numbering: (1) mean air temperature of the day before chlorophyll fluorescence measurements, (2) mean PAR of the day before measurements, (3) mean RH of the day before measurement, (4) maximum air temperature of the day before measurements, (5) maximum RH of the day before measurements, (6) minimum air temperature of the day before measurements, (7) minimum RH of the day before measurements, (8) mean air temperature of the period of measurement (7:00 to 11:00 a.m. of particular day), (9) mean PAR of the period of measurement, (10) mean RH of the period of measurement, (11) maximum air temperature of the period of measurement, (12) maximum PAR of the period of measurement, (13) maximum RH of the period of measurement, (14) minimum air temperature of the period of measurement, (15) minimum PAR of the period of measurement and (16) minimum RH of the period of measurement.

| Dependent Parameter | Climatic Parameter (Spearman Correlation, $p < 0.05$) | | | | | | | |
|---|---|---|---|---|---|---|---|---|
| | **1** | **2** | **3** | **4** | **5** | **6** | **7** | **8** |
| $F_V/F_M$ | −0.341 | −0.050 | −0.111 | −0.457 | −0.078 | −0.332 | −0.025 | −0.553 |
| $PI_{ABS}$ | −0.351 | −0.101 | −0.071 | −0.537 | −0.005 | −0.285 | 0.031 | −0.599 |
| $ET_0/RC$ | −0.152 | −0.276 | 0.166 | −0.299 | 0.268 | 0.083 | 0.413 | −0.110 |
| $DI_0/RC$ | 0.379 | 0.057 | 0.099 | 0.513 | 0.088 | 0.333 | 0.066 | 0.643 |
| | **9** | **10** | **11** | **12** | **13** | **14** | **15** | **16** |
| $F_V/F_M$ | −0.169 | 0.119 | −0.187 | −0.215 | 0.250 | −0.544 | −0.162 | 0.049 |
| $PI_{ABS}$ | −0.253 | 0.115 | −0.296 | −0.271 | 0.149 | −0.493 | −0.247 | 0.103 |
| $ET_0/RC$ | −0.325 | 0.263 | −0.201 | −0.341 | 0.051 | 0.088 | −0.227 | 0.375 |
| $DI_0/RC$ | 0.180 | −0.086 | 0.268 | 0.211 | −0.206 | 0.592 | 0.215 | −0.019 |

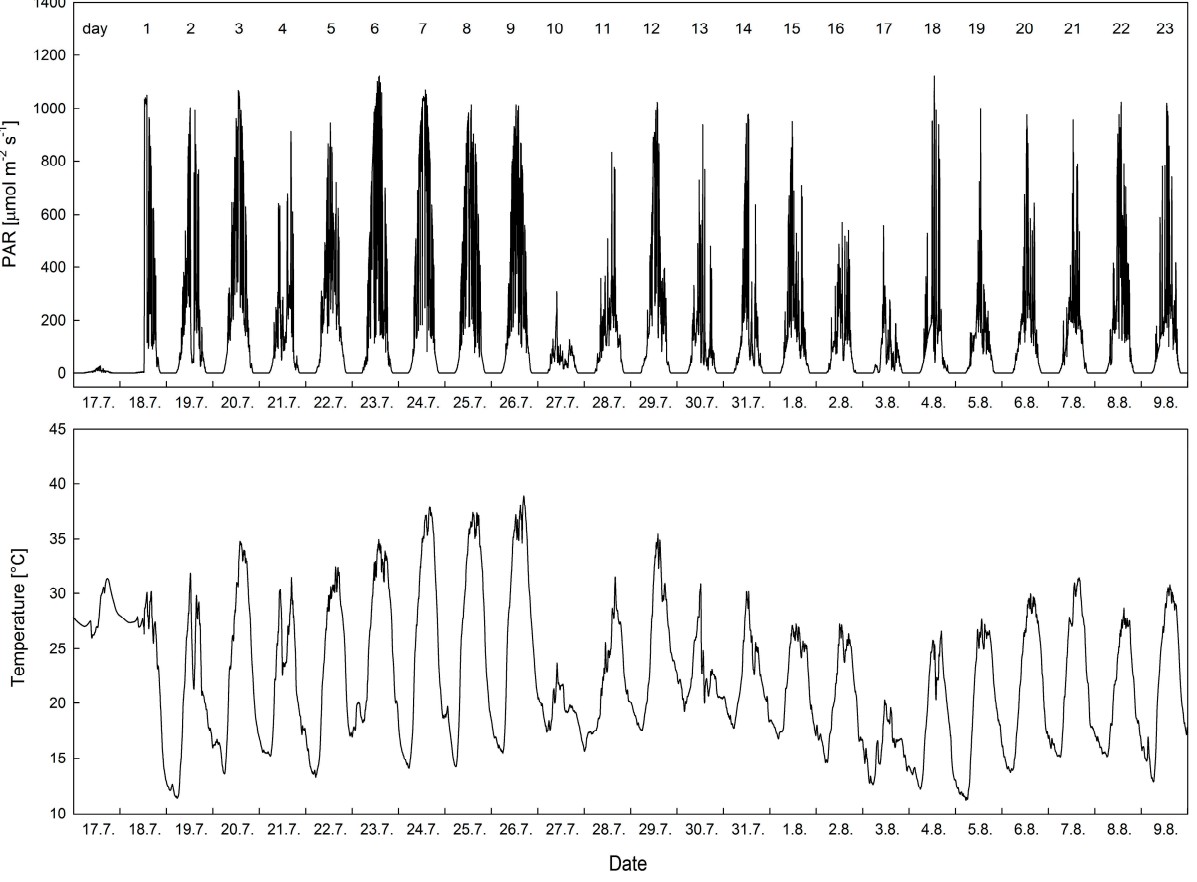

**Figure 7.** Daily courses of photosynthetically active radiation (PAR) and air temperature measured at the height of 1.5 m in the maize stand within the period of investigation (17 July to 9 August).

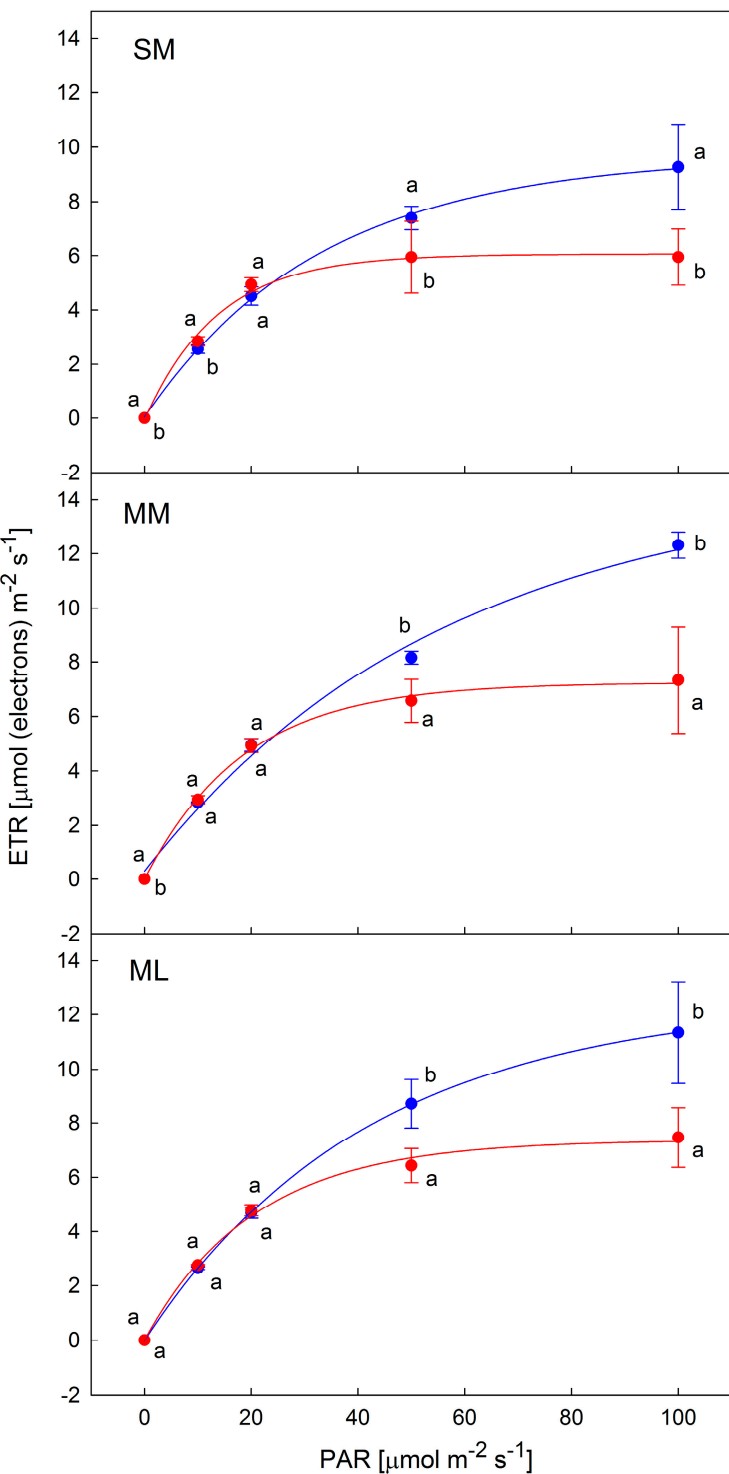

**Figure 8.** Light response curves of the photosynthetic electron transport rate (ETR) evaluated for the best (blue symbols) and worst days (red symbols). The best days were day 14 (SM), day 12 (MM) and day 12 (ML). The worst days were day 21 (SM), day 23 (MM) and day 19 (ML). Data points represent means of five replicates $\pm$ standard deviations. Characters (a, b) indicate statistically significant differences.

Above-ground biomass expressed in dry mass per hectare reached the values presented in Table 4. When considering maize biomass, it reached the highest value in SM, followed by the ML and MM intercropping systems. When considering overall biomass

(including maize plus particular intercropping species), it was highest in the MM, followed by ML an SM treatments.

**Table 4.** Above-ground biomass for *Z. mays, M. albus,* and *Lupinus* sp. estimated for sole maize (SM) and the two intercropping systems maize with white sweetclover (MM) and *Lupinus* sp. (ML).

| System | *Z. mays* [kg (DM) ha$^{-1}$] | *M. albus* (MM) *Lupinus* sp. (ML) [kg (DM) ha$^{-1}$] | Overall Productivity [kg (DM) ha$^{-1}$] |
|---|---|---|---|
| Sole maize (SM) | $15\,676 \pm 508$ | | $15\,676 \pm 508$ |
| Intercropping (MM) | $14\,514 \pm 874$ | $1\,381 \pm 102$ | $15\,895 \pm 521$ |
| Intercropping (ML) | $14\,806 \pm 1356$ | $1\,045 \pm 221$ | $15\,851 \pm 699$ |

## 4. Discussion

### 4.1. Microclimate

The effects of the microclimate on the parameters related to the primary processes of photosynthesis in *Zea mays* were apparent mainly during the episodic warm weather (days 6–9) and the other period (days 20–25, i.e., 4–9 July) marked by a gradual increase in air temperature. The first period caused a significant decrease in the chlorophyll fluorescence parameters derived from OJIP, indicating a strong limitation of primary photosynthetic processes. The latter period caused a limitation of parameters derived from ETR light response curves.

### 4.2. Photosynthetic Parameters Derived from OJIPs

Intercropping of maize with both *Lupinus* sp. and *Melilotus albus* brought about an increase in PI$_{ABS}$ and ET$_0$/RC, which can be explained as an intercropping species-dependent increase in energy flow through PSII and thylakoidal electron carriers in chloroplast. In this concept, an increased ATP and NADPH formation and their utilization in the biochemical processes of photosynthesis might be expected. Effective utilization of absorbed light energy in PSII and the (photosynthetic) linear electron flow in maize with MM treatment might be supported by the fact that thermal dissipation (DI$_0$/RC) reached lower values in the MM than in the SM treatment.

Time courses of the OJIP-derived chlorophyll fluorescence parameters revealed a markable decrease of F$_V$/F$_M$, PI$_{ABS}$ and ET$_0$/RC found on day 9. This decrease might be attributed to a limitation of PSII function, caused by two stress factors—high air temperature (over 35 °C—see Figure 7) accompanied by low RH (below 20%). The air temperature and PAR of the day before the day of measurements had a similar role in the limitation of the above OJIP-derived parameters. The effects (on OJIPs) of air temperature and PAR recorded for the period of measurement were not that apparent. This might be attributed to the fact that the measurements were done before 11:00 local time. Therefore, the plants did not witness the midday, and possibly even afternoon, depression of photosynthesis on the day of measurements. However, direct effects of air temperature on the day of measurements were also apparent. A high temperature-induced decrease in F$_V$/F$_M$, PI$_{ABS}$ and ET$_0$/RC found on day 9 is consistent with the evidence reported for plants treated by high temperatures [23] and have been interpreted as a direct effect of high temperature on inhibition of PSII. These negative changes to PSII functioning are typically reflected in a high temperature-induced decrease of chlorophyll fluorescence, which is demonstrated as a flattening of the OJIP curves [24] and an increase in V$_J$ chlorophyll fluorescence signal (see Figure 4). Apart from temperature, low RH is reported to cause ET$_0$/RC decline [25].

In the intercropping system of maize with *Lupinus* sp. and *Melilotus albus*, the microclimate of the stands was affected mainly by the shading effects of maize. Maize plants were always taller than the other two intercrop species, which resulted in a high absorption of incident PAR and, therefore, the limited availability of PAR to the two species (data not shown). The change in microclimate caused by the maize canopy is reported from

several intercropping systems. Liu and Song (2012) [26] refer to absorption higher than 50%. Consequently, changes in air temperature and relative air humidity appeared in the lower part of the *Zea mays* canopy, as well as in the canopy of *Lupinus* sp. and *Melilotus albus* in the ML and MM systems. Such a phenomenon is well documented, for example, in maize-soybean intercropping systems [27] and is dependent mainly on dimensions, growth rate of the target species (maize), row spacing and their azimuthal orientation [28]. In our study, PAR absorption by maize rows caused a light microclimate good enough for the successful growth and development of *Lupinus* sp. and *Melilotus albus,* which affected the microclimate (T and RH in particular) of the lower layer of the maize canopy. These changes were beneficial for maize photosynthesis and growth since the majority of the OJIP-derived parameters related to PSII functioning showed higher values for ML and MM systems than for maize alone. The intercropping-induced increase in the photosynthetic performance of maize has also been reported by [9] and attributed to a higher water use efficiency of maize cultivated in an intercropping system. Similarly, ref. [29] brought the evidence of the higher photosynthetic performance of maize when intercropped with soybeans, which is consistent with our results. However, it still remains an open question whether or not this is a consequence of the water efficiency of maize. Both a lower [30] and higher [31] lack of complementary water use in intercropped maize has been reported. The effect of the MM system on $F_0$ decrease in maize plants (compared to SM and ML) remains unknown, however, it might be attributed to the generally good performance of primary photosynthetic processes in the PSII of the MM plants.

*4.3. Light Response Curves of ETR*

Our data indicated the negative effect of air temperature on photosynthetic ETR curves. The best curves, achieved at optimum leaf temperature, (see Figure 8) showed somewhat higher ETR values for *Zea mays* with white sweetclover (MM) than the other two (*Zea mays* with *Lupinus* sp. (ML). and *Zea mays* alone (SM)). This suggests that the photosynthetic linear electron transport rate works more efficiently for *Zea mays* with *Melilotus albus*. At high temperatures (days 6–8, Figure 7), ETR values were found to be lower in the light response curves. This might be attributed to the fact that the primary photochemical reactions in PSII, as well as the carbon fixation rate in the stroma of chloroplasts, have been reported to be the primary sites of injury [32]. Therefore, due to the high-temperature-induced inactivation of PSII, photosynthetic activity was inhibited and ETR reduced [33]. Furthermore, due to the high-temperature-induced decrease in the activity of Rubisco, $CO_2$ metabolism is reduced as well. These metabolic changes result in a decrease in photosynthetic capacity and photochemical efficiency [34,35]. Such an interpretation is supported by evidence from laboratory-based experiments that showed chlorophyll fluorescence-based ETR decline in plants treated by temperatures above the photosynthetic optimum [36]. Recently, attempts were made to evaluate the light response curves of the net photosynthetic $CO_2$ assimilation rate from ETR data [37], with emphasis given to the effect of high leaf/air temperature on the limitation of photosynthesis. Lowest ETR values (worst curves, Figure 8) were found, however, on the day with a relatively low air temperature, i.e., August 4th to 9th (i.e., days 20–25 of the experiment). This indicates a limitation in photosynthetic performance by another co-acting factor. The most limiting factor for ETR was soil moisture, which decreased dramatically from the 4th to the 9th (data not shown). A decrease in soil humidity typically leads to drought stress in plants and the severe limitation of photosynthesis due to decreased stomatal conductance, as shown earlier for numerous crops [14] including maize [38–40].

In spite of significant positive differences found in photosynthetic parameters in *Z. mays* intercropped with legumes, our data did not show any statistically significant increase in biomass production and grain yield of *Z. mays*. If an assumption of a positive relation between primary photochemical photosynthetic processes (measured in our study) and biochemical processes related to $CO_2$ fixation is made, then, a question arises where the extra assimilates are utilized or allocated in *Z. mays.* Similar study with sole maize and

maize–legume intercropping systems [41] found that the intercropping legume did not affect maize growth and biomass. The authors reported even decreased biomass and grain production in *Z. mays* in the intercropping system with a legume. This is in agreement with the review of [42] which reported that the advantages of monocropping yield benefits over the maize/legume can be attributed to the interspecific competition for space, nutrients, water and light.

## 5. Conclusions

Maize row-intercropping systems with the two flowering legume plants tested in our experiment have proven that the use of *Lupinus* sp. and *Melilotus albus* as intercrops may be an alternative to sole maize cropping. It might be supported by the fact that *Zea mays* photosynthetic parameters related to the performance of PSII reached higher values in intercropping systems with *Lupinus* sp. and *Melilotus albus* than in a maize monoculture. This is consistent with the recent study of [41] that stated higher net photosynthesis *Z. mays* intercropped with a cowpea (legume) than in sole maize. It might also be concluded that the two cropping partners can mitigate negative environmental impacts and even create a microclimate beneficial for *Zea mays* photosynthesis and productivity. Additionally, the two cropping partners have a positive effect on soil sustainability, since they protect the agroecosystem from erosion. Therefore, legumes are considered as prospective plants for soil conservation systems used in future agricultural practices in order to improve soil quality properties [43].

**Author Contributions:** Conceptualization—J.L., A.K., J.E. and P.V.; methodology—J.L. and P.V.; investigation—J.L., M.B. and J.H.; data curation—J.L., P.V., B.Z. and J.E.; writing, review and editing—M.B. and J.H.; visualization—A.K., J.E. and J.H. All authors have read and agreed to the published version of the manuscript.

**Funding:** The study referred to in this paper was supported by the framework of the Institutional Support for Long-Term Conceptual Development of a Research Organization (DKRVO), reg. no. MZE-RO1722, funded by the Ministry of Agriculture of the Czech Republic.

**Data Availability Statement:** Data available upon request directed to the main author (Jaroslav Lang, lang@vupt.cz).

**Acknowledgments:** The authors express their thanks to the above-specified funding that supported the experiments referred to in this study.

**Conflicts of Interest:** The authors declare no conflict of interest.

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
