# Peer review of "Stimulative Effects of Lupinus sp. and Melilotus albus Underseed on the Photosynthetic Performance of Maize (Zea mays) in Two Intercropping Systems"

_agronomy, doi:10.3390/agronomy13010163_

Round 1
Reviewer 1 Report
English is to be improved.
Both the intercropping plants are lguminaceous and fix atmospheric nitrogen to soil. Is this the effect on photosynthesis?
If yes rewrite the paper accordingly.
If NO, rewrite accordingly.
What is the objective? It is not clear in the Introduction. Also, no mention is given regarding thepurpose of the present study.
When all the fluorescence measurements were taken? Time of the day?
Repeatition of work on soybean-maize studies by Guo et al.
Authors can explain a mechanism for the interaction, how the soil moisture or other factors might be affecting the photosynthetic performance in MM intercrop.
The paper needs major revision.
Author Response
ad attachment

Reviewer 2 Report
Manuscript "Stimulative Effects of Lupinus sp. and Melilotus officinalis Underseed on the Photosynthetic Performance of Maize (Zea mays) in Two Intercropping Systems” is devoted to the study of the photosynthesis activity of Maize under various sowing options. The authors recorded the parameters of photosynthesis activity with small time intervals (2-3 days) during the summer months. Such field measurements are of undoubted interest. I especially want to note that the authors performed a very correct and accurate measurement of environmental conditions. In combination with a non-invasive method of recording photosynthesis, this makes it possible to make a connection between the activity of photosynthesis and environmental conditions.
However, in its current form, the manuscript meets a number of questions.
1. The authors claim that co-cultivation gives a better result in comparison with the cultivation of monoculture (Title, Abstract, Conclusion). However, it does not follow from the presented results that there are statistically significant differences between the experimental groups. In the absence of differences, one group cannot be said to be better than the other. Based on this, the conclusions presented in the work are incorrect.
2. What is the yield of each group?
3. The authors explain the changes in the activity of photosynthesis by the influence of factors such as temperature and humidity. However, correlation analysis of photosynthesis activity parameters against temperature and humidity has not been performed. There is a complete set of values for all parameters. Why was the analysis of the dependence of ETR on T, ETR on H not performed?
4. In the Materials and Methods section, there is no description of the number of plants measured in each group. There is also no information about what the bars on the charts represent - SE, SD? What statistical methods were used to compare different treatment groups?
5. Measurements of the parameters of photosystem 2 are quite long. First, dark adaptation for 15 minutes, then 4 light intensities of at least 1 minute each. The measurement start time (morning) is indicated in the Materials and Methods section. There are 3 experimental groups, in each of the groups a certain number of plants were measured. I hope not less than 5. In total, when using one device, the measurement time is 0.3x5x3=5 hours. In the first plants, the parameters of photosynthesis will be determined, indeed, in the morning hours, but in the latter - in the daytime. In the morning and afternoon hours, Fv/Fm will not differ, but the effective quantum yield of photosystem 2 can differ quite significantly in the afternoon and morning hours, as evidenced by literature data. This can present a serious problem in the correctness of the measurements.
(A man stood motionless for 20 minutes at each leaf? The handheld instrument does not include a tripod. This question is just for me)
6. Why and when does the decrease in Fv/Fm occur? Is there damage to the photosynthetic apparatus? Were Fv/Fm lowest on the day with the highest temperature or the next day? The text of the manuscript says that the minimum Fv / Fm is on the day with the maximum temperature. If the measurements were made in the morning hours, and the maximum temperature occurs at noon, the temperature of the current day will not yet have an effect on the photosynthetic apparatus. What caused the damage?
7. Light curves were obtained every day of measurements. The graph shows the best and worst days. But it is not even stated whether these days are the same for all groups. Figure 8 allows you to compare the minimum and maximum values of ETR within the same experimental groups. But it does not allow comparing different experimental groups with each other. To do this, all 3 options must be presented on the same graph. For example, you can make two charts - the best day - 3 curves, the worst day - 3 curves. But even the optimization of this figure does not answer the question of what this figure demonstrates. Dynamics of ETR at maximum illumination can be given.
8. Section Discussion is not based on the presented results. The missing differences are discussed.
9. Dates are shown on the diagrams of temperature, illumination, and other graphs show the days of the experiment. The x-axes of the first graphs should also contain the days of the experiment along with the dates.
10. The text of the manuscript does not display the symbol F in FPSII (signature of figure 7, table 2, etc.)
Author Response
ad attachment
